# Health Literacy is Associated with Health Behaviors in Students from Vocational Education and Training Schools: A Danish Population-Based Survey

**DOI:** 10.3390/ijerph17020671

**Published:** 2020-01-20

**Authors:** Charlotte Demant Klinker, Anna Aaby, Lene Winther Ringgaard, Anneke Vang Hjort, Melanie Hawkins, Helle Terkildsen Maindal

**Affiliations:** 1Steno Diabetes Center Copenhagen, Health Promotion, Niels Steensens vej 6, 2820 Gentofte, Denmark; 2Department of Public Health, Bartholins Alle 2, Aarhus University, 8000 Aarhus, Denmark; 3School of Health and Social Development, Deakin University, Geelong, VIC 3220, Australia

**Keywords:** health literacy, health equity, health promotion, health behavior, low socio-economic status, vocational education and training, quality of life

## Abstract

Health literacy has been identified as an important and changeable intermediary determinant of health equity. Vocational education and training (VET) schools are a relevant setting for health behavior interventions seeking to diminish health inequities because many VET students come from low socio-economic status backgrounds. This study examines VET students’ health literacy and its association with health behavior based on a cross-sectional survey among 6119 students from 58 VET schools in Denmark in 2019. Two scales from the Health Literacy Questionnaire was used to assess domains of health literacy. Data were analyzed using Anova and logistic regression. The study population consisted of 43.4% female, and mean age was 24.2 years (range 15.8–64.0). The health literacy domain ‘Actively managing my health’ mean was 2.51, SD 0.66, and ‘Appraisal of health information’ mean was 2.37, SD 0.65. For both domains, being female, older age, attending the VET educational program Care-health-pedagogy, and higher self-rated health were associated with higher scale scores. In the adjusted analyses, lower scale scores were associated with less frequent breakfast, daily smoking, high-risk alcohol behavior and moderate-to-low physical activity. Our results show that low health literacy is associated with unhealthy behaviors in this population. Our results support and inform health literacy research and practice in educational institutions and services.

## 1. Introduction

Health behavior among adolescents and young adults affects life expectancy as well as health and wellbeing throughout the life course [1]. Exploring predictors and modifiers of youth health behavior is therefore a crucial step towards effective early prevention of poor health outcomes. Low socio-economic status (SES) has been identified as such a predictor [2,3], and one possible mediating factor is health literacy [4,5,6].

Health literacy can be defined as “the combination of personal competencies and situational resources needed for people to access, understand, appraise and use information and services to make decisions about health. It includes the capacity to communicate, assert and act upon these decisions” [7]. Consequently, the concept of health literacy constitutes individual competencies and capacities but is exercised in a dynamic interplay with available resources [8].

Health literacy has been associated with many social health determinants including high educational attainment, high income level, high self-reported social status, and cohabitation [9,10,11]. However, the evidence in young adults is scarce. A recent review indicates that health literacy may act as a modifier between SES and health outcomes [12], placing health literacy as an important dynamic and changeable intermediary determinant of health equity. The review includes only one study in younger adults examining the role of health literacy in relation to asthma outcomes in ethnic minority groups [13]. In a review from 2016, Sansom-Daly et al. explored the association between health literacy and health behavior in people aged 10–39 years. They found that lower health literacy scores were associated with obesity, smoking, and poor self-reported nutrition behaviors [14]. Other reviews have confirmed possible associations between health literacy and different health behaviors in adolescents [15,16], although other determinants, such as psychosocial factors, may also play a large role alongside health literacy for the health outcomes of young adults [17].

The focus of the present study is on students enrolled in Danish Vocational Education and Training (VET) schools. Compared with high school students, Danish VET students seem to have more pronounced risk behaviors in terms of poor diet, frequent smoking, low levels of physical activity [18,19], and high school dropout rates [20]. Further, many students attending VET come from families of low SES [21,22]. As such, VET schools are an important setting for health promotion [23,24], which could mitigate the effect of social inequity in health. The multi-dimensional and modifiable construct of health literacy could form a foundation for such health promotion efforts to improve individual and population health outcomes [25,26,27].

In a study from 2018, Rademakers et al. distinguish between cognitive and non-cognitive aspects of health literacy (i.e., ‘capacity to think’ and ‘capacity to act’) but shows how they are mutually enhancing, and how they both are crucial in activities aimed at improving self-management and health behavior [28]. However, few studies measure health literacy comprehensively enough to make that distinction, and even fewer measure it in a young and presumable low SES population. The Health Literacy Questionnaire (HLQ) consists of nine scales, each of which measure a domain of health literacy, and collectively cover both cognitive and non-cognitive capacities [26]. Two scales that relate to the ‘capacity to act’ (Scale 3. Actively managing my health) and ‘capacity to think’ (Scale 5. Appraisal of health information) were included in this study. To our knowledge, this is the first time these two HLQ scales have been used in a VET school population.

The purpose of this study was twofold. The first aim was to examine VET students’ health literacy within population subgroups. The second aim was to investigate the association between health behaviors of Danish VET students and health literacy.

## 2. Materials and Methods

### 2.1. Study Setting, Design and Data Collection

In Denmark, 23% of all primary- and lower secondary school students continue into VET schools, which include more than 100 different educations (e.g., agricultural, commercial, technical, or social healthcare educations) [29]. The 100+ different educations are compiled by the Danish Ministry of Education into four main educational programs: (1) Care, health and pedagogy, (2) Administration, commerce and business service, (3) Food, agriculture and hospitality, and (4) Technology, construction and transportation. There is a large variation in the age of VET students. A large part of students continue directly from primary and lower secondary school to a VET program (age 15–17 years old), while others enroll later in adult life [30]. VET varies in length from two to five-and-a-half year with an average length of four years. The program combines alternating school-based training and workplace-based training, with approximately one-third of the time being school-based training [21,31]. A Danish VET reform (2015) introduced two educational VET levels: VET-normal and VET-higher. VET-normal qualifies the student to work within a specific trade (e.g., carpentry or hairdressing), while VET-higher qualifies for a specific trade *and* further education (e.g., university) because students also achieve a high school diploma.

This cross-sectional study is based on data from the Danish Health and Wellbeing survey in VET schools 2019 (Health-VET 2019) describing health and wellbeing among a national sample of VET students. Data were collected from students enrolled in Danish VET schools aiming of an approximate sample size of 5% of the student population within each main educational program and across the five Danish regions. To ensure a representative sample, for each region, we included at least two schools offering each of the four VET educational programs (e.g., a minimum of eight schools per region). Schools within main educational program and region were randomly ordered, then invited to participate and, if they consented, included. If a school declined to participate, the next school from the randomly ordered list, matching region and main educational program, was invited to take part in the study, until an approximate sample size of 5% was reached.

Data collection took place during school hours between February and June 2019 using electronic questionnaires. Researchers were present during data collection to explain the purpose of the study and to give practical support. The entire data collection process took approximately 45 min per school class and it took an average of 30 min for the students to fill out the questionnaire.

### 2.2. Measures

The questionnaire contained questions about health literacy, diet, smoking, substance abuse, alcohol, physical activity, sleeping and digital technology use, mental health, and general health. The following measures were used in this study.

#### 2.2.1. Health Literacy

The HLQ was developed using a validity-driven approach [32] and consists of 44 items that measure nine domains of health literacy [26]. Validity testing was initially conducted in Australia where it showed strong psychometric properties [26,33,34]. It has been translated to 30 languages and is being used in more than 50 countries with published studies indicating that the psychometric properties remain strong in other languages and cultures [9,35,36,37,38]. The HLQ was translated to Danish by Maindal et al. [37]. Due to space restrictions on the questionnaire two out of the nine scales from the Danish HLQ were included in the questionnaire and used in this study: Scale 3. Actively managing my health, and Scale 5. Appraisal of health information. Each of the two scales consist of five items with response options on a four-point scale: strongly disagree/disagree/agree/strongly agree. HLQ scale scores were calculated separately for each scale as the mean score of the number of items answered. The setup of the electronic questionnaire did not allow missing values. See Table 1 for descriptions of high and low scoring for these two scales.

#### 2.2.2. Health Behaviors

Four measures of health behavior were included: smoking, diet, alcohol and physical activity.

Smoking was assessed using the question ‘Do you smoke cigarettes? (not electronic cigarettes)’ and categorized as daily smoker (smoking daily), occasional smoker (smoking weekly/smoking less than once a week), former smoker (used to smoke/have tried smoking), and never smoker (never smoked—not even one puff) [39].

Dietary behavior was measured as frequency of having breakfast during weekdays as this has been found to correlate with both disease risk [40] and obesity [41]. It was categorized as always/sometimes (1–4 times during weekdays) and seldom/never (less than once a week or never).

Alcohol behavior was measured as high/intermediate/low risk based on the Danish Health Authority recommendations [42]. A weekly drinking pattern above 14/21 units for females/males, respectively was considered high, above 7/14 but less than 14/21 as intermediate risk, and not drinking at all or drinking below or 7/14 units per week was categorized as no/low risk.

Physical activity (PA) was measured as adherence to the World Health Organization (WHO) minimum (WHO-min) and health enhancing (WHO-HE) recommendations for physical activity [43] using the Nordic Physical Activity Questionnaire-short (NPAQ-short) [44]. NPAQ-short comprises two open-ended questions about weekly moderate-to-vigorous physical activity and vigorous physical activity. PA behavior was categorized as high PA (adheres to WHO-HE/being physically active for at least 300 min of moderate activity or 150 min of vigorous activity per week), moderate PA (adheres to WHO-min/being physically active for at least 150 min of moderate activity or 75 min of vigorous activity per week) and low PA (does not adhere to WHO-min or WHO-HE).

#### 2.2.3. Health Status; Socio-Demographic and Educational Factors

Two measures of health status were included: self-rated health and body mass index (BMI). Self-rated health was measured using one item: ‘In general, would you say your health is…’ with the five response options aggregated into three categories: excellent/very good, good, fair/poor. BMI (weight/(height)^2^) was calculated using self-reported weight (kg) and height (m) and classified as underweight (BMI < 18.5), normal (BMI 18.5 – <25), overweight (BMI 25 – <30), and obese (BMI 30+).

Sex and age were included as socio-demographic factors. Information on age and sex were extracted from the personal unique Social Security Number assigned to all people in Denmark (97.8%) or based on self-report (2.2%). Age was split into three categories: 15–18, 19–25 and 26+ years.

Educational VET level (VET-normal or VET-higher) and the four main educational programs were included as educational factors and based on self-report. Only educational level (VET-normal compared to VET-higher) was included as a measure of SES in the analysis as the population consists of students. For most students, a VET degree will lead to manual or trade work, which is classified as a short education, and this is generally considered a low SES group. Exceptions to this may be students attending VET-higher.

### 2.3. Statistical Analysis

One-way ANOVAs were used to assess associations between the two HLQ scales and health status, socio-demographic and educational factors. Associations between the two HLQ scales and health behaviors were analyzed using logistic regressions with each health behavior as the dependent variable and each HLQ scale as the predictor variable. Health behaviors with more than two categories were analyzed using dummy variables with the most favorable health behavior as the reference category. Unadjusted odds ratios (OR crude) and adjusted odds ratios (OR adjusted) with a 95% confidence interval (CI) were reported. Odds ratios were adjusted for health status, socio-demographic and educational factors (self-rated health, BMI, gender, age, VET level and main educational program). *p* values < 0.1 (two-tailed) was considered a trend, and *p* < 0.05 was considered statistically significant. A sole focus on a traditional level such as 0.05 can fail to detect important associations [45,46]. All statistical analyses were performed using IBM SPSS Statistics v.25 (Armonk, NY, USA).

### 2.4. Ethics and Approval

This study was approved by the Capital Region of Denmark (VD-2018-485 ref. no 6743) and adheres to all GDPR-regulations. It was explained to all study participants that participation in the study was voluntary and that all information collected would remain confidential. All information was given both orally and in writing prior to data collection. Participants between 15 and 18 years old were further provided with a letter to take home to inform their parents about their participation in the study. Legal age to consent to participate in studies without parental consent is 15 years old according to the Danish Health Law. Digital informed consent was obtained from all participants using two-factor authorization to ensure proper identification. Also, the students were made aware that they could withdraw their consent at anytime, and information on how to do this was provided both orally and in writing (phone and email).

## 3. Results

### 3.1. Sample Characteristics

A total of 85 schools were invited to take part in the study, with 58 (68.2%) agreeing to participate in Health-Vet 2019. A total of 6119 students participated, which corresponds to 5.9% of all students enrolled at VET schools in Denmark.

Characteristics of the respondents are presented in Table 2. The population was characterized by most respondents being 25 years or younger of age (73.2%, mean age 24.2 (range 15.8–64.0)) and a little more than half (56.6%) were males. Most respondents were enrolled at VET-normal (80.8%) with the largest main educational program being Technology-construction-transportation (40.3%).

Regarding health status and health behaviors, approximately one in seven (13.5%) respondents reported fair or poor health status while two in five (39.9%) were overweight or obese. Only 46.7% had breakfast every day, 28.9% were daily smokers, 17.7% had a high-risk alcohol consumption, and 29.7% reported low physical activity levels.

### 3.2. Health Literacy and Socio-Demographic and Educational Factors, and Health Status

Cronbach’s alpha and composite reliability were found to be high in both scales: 0.87/0.85 for Scale 3 and 0.84/0.83 for Scale 5, respectively. In Table 3, the health literacy scale scores have been reported in sub-populations based on socio-demographic and educational factors and health status characteristics. We found statistically significant differences in both scales by sex (lowest scores in males), age-groups (highest scores in the >= 26 years old), main educational program (highest in students attending Care-health-pedagogy), and self-rated health (lowest in the group reporting fair/poor health). For educational level, only a significant difference was found for Scale 3 (lowest in people attending VET-normal). No differences in health literacy were found between groups with different BMI scores.

### 3.3. Health Literacy and Health Behaviour

Associations between the scale scores and health behaviors are reported in Table 4 for each scale separately. In the adjusted analyses we found that higher scores in Scale 3. Actively managing my health significantly decreased the odds of having breakfast somedays (adjusted OR 0.72 (CI 0.65–0.79)), seldom or never having breakfast (adjusted OR 0.59 (CI 0.53–0.65)), daily smoking (adjusted OR 0.64 (CI 0.57–0.72)), high risk alcohol consumption (adjusted OR 0.78 (CI 0.69–0.87)), moderate physical activity (adjusted OR 0.66 (CI 0.59–0.76)), and low physical activity (adjusted OR 0.40 (CI 0.36–0.45)). Likewise, we found, that higher scores for Scale 5. Appraisal of health information significantly decreased the odds of seldom or never having breakfast (adjusted OR 0.76 (CI 0.68–0.84)), daily smoking (adjusted OR 0.82 (CI 0.73–0.92)), moderate physical activity (adjusted OR 0.84 (CI 0.74–0.95)), and low physical activity (adjusted OR 0.57 (CI 0.51–0.64)).

## 4. Discussion

The aim of this study was twofold: (1) to describe health literacy levels in two health literacy domains within population subgroups based on socio-demographic, and educational factors, and health status and, (2) to investigate the association between health behaviors of VET students and two domains of health literacy. We found the lowest health literacy scores were among males, the youngest students, students with low self-reported health, and students attending other VET education programs than care-health-pedagogy. These may be possible target groups for health promotion interventions to improve capacity to manage health and critically appraise health information, and perhaps reduce health inequities. Our results demonstrate that, for active management of health and critical appraisal of health information, VET students with higher health literacy scores were systematically and consistently associated with a decreased odds of multiple unhealthy behaviors. These results indicate that health literacy may be an important determinant of health behaviors in this population.

### 4.1. Interpretations

Using a variety of measures, many studies have found a range of aspects of health literacy to be associated with socio-demographic and health related indicators [9,10,47,48]. Only some of these studies include young people and none of the studies include VET students. However, our study findings of low health literacy among males and the youngest students (Table 3) is partly supported by the literature, though the association may be dependent on the specific domains of health literacy in question [9,10,47].

Many studies confirm the association between self-reported health status and educational attainment [9,10,47] but, to the best of our knowledge, no study report on students at different educational levels and educational program in VET education. In accordance with our findings (Table 3), other studies have found positive associations between health literacy and self-reported health status in general populations, including younger adults [47,49,50], although the directionality of this association is still unsupported.

Most studies about the association between health literacy and health behaviors have been conducted in general populations or populations characterized by specific health conditions, such as heart disease or diabetes [47,48,51,52,53]. Like our findings (Table 4), study results are most consistent in the demonstration of positive associations between health literacy and physical activity and a healthy diet. The results are less clear for associations between health literacy and smoking and alcohol consumption. In young populations, the review by Sansom-Daly et al. reports that low health literacy may be associated with an unhealthy diet and smoking, but not physical activity levels [14]. The review includes a large Swiss study from 2013 where smoking, alcohol consumption, or cannabis use among young adults was associated with increased information seeking behavior and more extensive knowledge of risks [54]. A study of 2768 Danish Students (age 15–16 years) showed that higher alcohol intake was associated with gender, poor relationships with parents, negative or positive expectancies of the impact of alcohol, and the influence of peers and their alcohol use. The study suggests the development of alcohol-related health literacy skills as a way to diminish unhealthy alcohol consumption [53].

The study population in our study demonstrated fewer healthy behaviors than the general Danish population, as reported in the Danish Health Profile, which is conducted every fourth year [55]. For example, in the VET population the prevalence of daily smoking was 28.9% and a high-risk alcohol behavior was seen in 17.7% as compared to 20.5% daily smokers and 6.9% with a high-risk alcohol profile in the general Danish population. In relation to physical activity the VET population is only marginally worse off than the general population, with almost 1 out of 3 not adhering to the minimum requirements for physical activity (29.7% of the VET students do not adhere compared to 28.8% in the general population). Collectively, our results indicate the VET population may be at significant risk of developing non-communicable disease, e.g., Type 2 diabetes or coronary heart disease, later in life. Carefully designed health promotion interventions targeting this group is warranted. However, to our knowledge, only few interventions have been attempted with this group and with mixed results [56,57]. An important strategy to improve health behavior in VET students may be to improve health promotion capacity at the school level [58] as well as to develop and implement interventions that target student health literacy, both addressing ‘capacity to think’ and ‘capacity to act’.

### 4.2. Strength and Limitations

A strength of this study is the use of scales from the widely-used and extensively tested HLQ [35,36,37,59]. However, the validity of inferences derived from HLQ data from youth below the age of 18 has not been verified and so results for this age group must be interpreted with caution. A sensitivity analysis conducted leaving the youngest age group (15–18 years old) out of the analysis on the association between health literacy and health behavior did not change the conclusion of this study (Appendix A). This indicates that our results are robust, even though further research needs to be conducted to confirm the psychometric properties of the HLQ in young people.

A further strength of the study is the large sample size and that schools were invited to participate based on a random ordered list. The study sampled more than 5% of the total population within each educational program (5.1–9.7%) and each region (5.2–7.0%) except for the educational program Administration-commerce-business-and-service where 4.6% of all students participated. Thus, the sample is considered representative for Danish VET students present at the schools during the winter and spring semester. When a reason for non-participation at the school level was given, the main reason was lack of time or resources and it cannot be ruled out that non-participating schools may have a different student health and wellbeing profile. However, with a VET school participation rate of 68.2%, the representativeness of the target study population is considered high.

Also, this study is the first, to our knowledge, to adjust for and investigate differences between the four VET educational programs, as well as educational levels. Given the differences that were found, this study indicates that it is important not to consider VET students as one homogenous group but to observe that distinct differences between sub-populations (e.g., educational programs and levels) may exist, and need to be considered when planning health promotion interventions. Only a small proportion of students (3.4%) started the questionnaire but did not complete all questions, and therefore student compliance is considered high.

Our study has some limitations. As the study was based on a cross-sectional design, no conclusions about causality can be applied. Another limitation was that the measurements of health status and behaviors were self-reported and therefore subject to respondent bias. However, this is more likely to lead to an underestimation of effect sizes as less desirable health status and unhealthy health behaviors have been shown to be underreported using self-report [60,61]. Health behaviors are complex, and this study only included one measure for each behavior to indicate possible associations between health behaviors and health literacy. Future research to explore each health behavior in depth and its association to health literacy is needed.

## 5. Conclusions

This study demonstrates that higher health literacy scores among VET students were systematically and consistently associated with decreased odds of multiple unhealthy behaviors, indicating that health literacy may be an important determinant of health behaviors in this population. Based on our results, health promotion activities to support students to manage their health and critically appraise health information would be best targeted at male students, younger students, those with low self-reported health, and those who attend particular VET education programs. These were the students with the lowest health literacy scores and health behaviors and therefore most likely to have the poorest health outcomes in later life. This study advances health literacy research in VET students, and informs VET, and potentially other educational institutions, about direction of interventions to support healthy behaviors and equitable health outcomes among VET students later in life.

## Figures and Tables

**Table 1 ijerph-17-00671-t001:** Descriptions of high and low scoring for two health literacy domains.

**Scale 3. Actively Managing My Health**
**High**: Individuals with high scores recognize the importance of taking responsibility for their own health. They proactively engage in their own care and make their own decisions about their health.
**Low**: Individuals with low scores do not see their health as their responsibility. They are not engaged in their health care and they regard health care as something that is done to them.
**Scale 5. Appraisal of Health Information**
**High**: Individuals with high scores are able to identify good information and reliable sources of information. They can resolve conflicting information by themselves or with help from others.
**Low**: Individuals with low scores cannot understand most health information no matter how hard they try. They get confused when there is conflicting information.

**Table 2 ijerph-17-00671-t002:** Characteristics of socio-demographic and educational factors, health status, and health behavior among VET school students (n = 6119 − 5148 *).

	Categories	N *	%
Sex	Male	3449	56.6
	Female	2647	43.4
Age (mean 24.2, range 15.8–64.0)	15–<19 years	2340	39.0
	19–<26 years	2048	34.2
	26+ years	1608	26.8
Educational level	VET-normal	4945	80.8
	VET-Higher	1174	19.2
Main Educational Program	Care, health and pedagogy	1763	28.8
	Administration, commerce and business service	1045	17.1
	Food, agriculture and hospitality	842	13.8
	Technology, construction and transportation	2469	40.3
Health status			
Self-rated health	Excellent/Very good	2653	43.2
	Good	2574	43.3
	Fair/poor	802	13.5
Body Mass Index	Underweight (<18.5)	380	6.4
	Normal weight (18.5–<25)	3169	53.7
	Overweight (25–<30)	1480	25.1
	Obese (30+)	871	14.8
Health behaviour			
Dietary habits (Breakfast during weekdays)	Every day	2803	46.7
	Somedays (1–4 days a week day)	1653	27.5
	Seldom or never	1547	25.8
Smoking status	Never smoker	1570	25.8
	Former smoker	2220	36.5
	Occasional smoker	531	8.7
	Daily smoker	1754	28.9
Alchohol intake	No-low risk	4344	71.9
	Intermediate risk	640	10.6
	High risk	1060	17.7
Physical activity	High level (Adhere til WHO-HE)	2677	50.9
	Moderate level (Adhere to WHO-min)	1025	19.5
	Low level (Do not adhere to WHO-min	1561	29.7

* N varies due to missing data. Abbreviations: WHO; World Health Organization, WHO-HE; WHO health enhancing requirements, WHO-min; WHO minimum requirements.

**Table 3 ijerph-17-00671-t003:** Health literacy domains by socio-demographic and educational factors, and health status among VET school students.

	Scale 3. Actively Managing Health (n = 5942)			Scale 5. Appraisal of Health Information (n = 5943)		
	Mean Score	SD	*p*-Value	Mean Score	SD	*p*-Value
Total sample	2.51	0.66		2.37	0.65	
Sex			0.000			0.000
Male	2.46	0.68		2.32	0.66	
Female	2.57	0.62		2.43	0.62	
Age			0.000			0.000
15-< 19 years	2.43	0.66		2.25	0.63	
19-< 26 years	2.49	0.64		2.38	0.64	
26+ years	2.65	0.63		2.54	0.63	
Educational level			0.042			0.259
VET-normal	2.50	0.66		2.38	0.65	
VET-higher	2.52	0.66		2.33	0.62	
Main Educational Program			0.000			0.000
Care, health and pedagogy	2.62	0.62		2.53	0.61	
Administration, commerce and business service	2.49	0.67		2.30	0.64	
Food, agriculture and hospitality	2.43	0.63		2.33	0.63	
Technology, construction and transportation	2.45	0.68		2.29	0.66	
Health status						
Self-rated health			0.000			0.000
Excellent/very good	2.61	0.72		2.40	0.70	
Good	2.48	0.57		2.37	0.59	
Fair/poor	2.26	0.65		2.25	0.65	
Body Mass Index			0.163			0.511
Underweight (<18.5)	2.39	0.71		2.26	0.71	
Normal weight (18.5–<25)	2.52	0.66		2.36	0.64	
Overweight (25–<30)	2.54	0.64		2.42	0.64	
Obese (30+)	2.44	0.61		2.37	0.65	

Abbriviations: SD; Standard Deviation.

**Table 4 ijerph-17-00671-t004:** Associations between health literacy domains and health behavior among VET-school students.

Health Behavior	Scale 3. Managing Health				Scale 5. Appraisal of Health Information			
	Crude		Adjusted *		Crude		Adjusted *	
	OR	(95% CI)	OR	(95% CI)	OR	(95% CI)	OR	(95% CI)
Dietary habits—breakfast								
Every day (ref)	1		1		1		1	
Somedays	**0.657**	**(0.597–0.724)**	**0.717**	**(0.647–0.794)**	**0.823**	**(0.747–0.906)**	0.907	(0.818–1.005)
Seldom or never	**0.534**	**(0.485–0.589)**	**0.588**	**(0.529–0.653)**	**0.690**	**(0.627–0.760)**	**0.755**	**(0.681–0.838)**
Smoking status								
Never (ref)	1		1		1		1	
Former	0.984	(0.889–1.088)	0.963	(0.864–1.073)	**0.879**	**(0.782–0.988)**	0.970	(0.870–1.081)
Occasional	0.811	(0.696–0.945)	*0.864*	>*(0.732–1.019)*	0.858	(0.734–1.003)	0.926	(0.783–1.096)
Daily	**0.632**	**(0.567–0.704)**	**0.642**	**(0.571–0.722)**	**0.830**	**(0.748–0.922)**	**0.816**	**(0.728–0.915)**
Alchohol								
No-low risk (ref)	1		1		1		1	
Intermediate risk	**0.860**	**(0.755–0.980)**	0.940	(0.817–1.082)	**0.844**	**(0.740–0.963)**	0.971	(0.842–1.119)
High risk	**0.649**	**(0.585–0.720)**	**0.777**	**(0.693–0.872)**	**0.688**	**(0.619–0.764)**	>*0.892*	>*(0.795–1.001)*
Physical activity								
High level (ref)	1		1		1		1	
Moderate level	**0.680**	**(0.604–0.765)**	**0.664**	**(0.585–0.755)**	**0.879**	**(0.782–0.988)**	**0.835**	**(0.737–0.947)**
Low level	**0.389**	**(0.350–0.433)**	**0.402**	**(0.359–0.451)**	**0.578**	**(0.523–0.640)**	**0.573**	**(0.514–0.639)**

* Adjusted for gender, age, educational level, main educational area, self-rated health, BMI. Bold: *p* < 0.05, Italic: *p* < 0.1. Abbriviations: OR; Odds Ratio, CI; Confidence Interval.

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
