# Peer review of "Health Literacy is Associated with Health Behaviors in Students from Vocational Education and Training Schools: A Danish Population-Based Survey"

_ijerph, 2020, doi:10.3390/ijerph17020671_

Round 1
Reviewer 1 Report
This is a well-written article examining how health behaviors are related to health literacy using data from the Danish population-based survey.
I have a few suggestions for the authors to consider:
Provide some justification for choosing two out of the nine scales from the Danish HLQ. Why not use all nine scales? Why did the authors choose these two scales in particular? In table 4, it'll be helpful to show all coefficients, including covariates. In table 4, the authors may consider adding information such as sample size. Have the authors considered data visualization? It's quite a bit of information in the article, using bar charts to present some of the findings may be helpful.Author Response
Reviewer 1 (BOLD)
Comments and Suggestions for Authors
Provide some justification for choosing two out of the nine scales from the Danish HLQ. Why not use all nine scales? Why did the authors choose these two scales in particular?
Thanks for this comment. In section 2.2.1 we have now added that only two out of nine scales were included due to space restrictions in the questionnaire. In the introduction, line 74-78 we explain that these two scales are used in this study as they relate to the ‘capacity to act’ (Scale 3. Actively managing my health) and ‘capacity to think’ (Scale 5. Appraisal of health information)
In table 4, it'll be helpful to show all coefficients, including covariates.
In table 4, the authors may consider adding information such as sample size.
As each adjusted result in table 4 is the result of a regression analysis, including specific sample sizes for each analysis, would complicate table 4 enormously. We have instead chosen to report the number of respondents for all variables used in the analyses in table 2 In table 4, we show all the coefficients that relate to the research question in play: what is the association between health behaviors of Danish VET students and health literacy, but not the coefficients for the covariates, as this does not provide further information on our research question.
Have the authors considered data visualization? It's quite a bit of information in the article, using bar charts to present some of the findings may be helpful.
We agree that the article presents quite a lot of data, and that the tables may seem heavy. However, we would like to maintain that the data presented is too complex to visualize as bar charts.
Reviewer 2 Report
The study entitled “Health literacy is associated with health behaviour in students from vocational education and training schools: A Danish population-based survey” concludes that low health literacy is associated with unhealthy behaviours in Danish population, being the health literacy an important determinant of health behaviors in this population.
The paper is well written and organized, but I do not recommend publication of the present version. I highlight minor issues that authors should address before the manuscript can be considered again for publication.
Introduction
The introduction section is quite complete and contains theoretical background. If appropriate, hypothesis could be considered.Material and Methods
More information about the psychometric properties of the measures used in the study is needed. As reliability varies in each test administration, it is necessary to report the reliability (for example, the alpha coefficient) obtained for the data at hand. Please, inform about this result for each instrument applied in Measures subsection.Discussion
Information about clinical and practical implications extracted from the study should be included.Author Response
Reviewer 3 (BOLD)
Introduction
The introduction section is quite complete and contains theoretical background. If appropriate, hypothesis could be considered.
Thanks for this comment. In the introduction we describe that lower levels of health literacy have been found to be associated with poorer health behaviors, and we prefer not to state a specific hypothesis statement
Material and Methods
More information about the psychometric properties of the measures used in the study is needed. As reliability varies in each test administration, it is necessary to report the reliability (for example, the alpha coefficient) obtained for the data at hand. Please, inform about this result for each instrument applied in Measures subsection.
We are reporting both Cronbach’s alpha and composite reliability for the two health literacy scales included, as this is the first time these scales have been used in a VET population. For the health behaviors and covariates included in the study we have chosen to include references to the psychometric properties of the measures (reliability and validity) where these are known. The included measures are categorical or continuous variables, and it is not possible to report the specific psychometric properties of these measures as this was not the aim of the more general questionnaire study
Discussion
Information about clinical and practical implications extracted from the study should be included.
Thank you for this comment. Due to this study taking place in VET schools it does not have any clinical implications per ser but some practical implications which we have highlighted in the conclusion. More specifically we state that health promotion activities would be best targeted at male students, younger students, those with low self-reported health, and those who attend particular VET education programs.